# Dissociation Behavior of Dislocations in Ice

**Takeo Hondoh** [1,2] 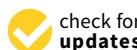

1    Institute of Low Temperature Science, Hokkaido University, N19W8, Sapporo 060-0819, Japan;
     hondoh@general.hokudai.ac.jp
2    Preset address: Professor emeritus at Hokkaido University, 2-2-107, Hassamu, Nishi-ku,
     Sapporo 063-0825, Japan

**Abstract:** Dislocations in ice behave very differently from those in other materials due to the very low energies of stacking faults in the ice basal plane. As a result, the dislocations dissociate on the basal plane, from a perfect dislocation into two partial dislocations with equilibrium width $w_e$ ranging from 20 to 500 nm, but what is the timescale to reach this dissociated state? Using physical models, we estimate this timescale by calculating two time-constants: the dissociation-completing time $t_d$ and the dissociation-beginning time $t_b$. These time constants are calculated for two Burgers vectors as a function of temperature. For perfect dislocations with Burgers vector <**c** + **a**>, $t_d$ is more than one month even at the melting temperature $T_M$, and it exceeds $10^3$ years below $-50\,°C$, meaning that the dissociation cannot be completed during deformation over laboratory timescales. However, in this case the beginning time $t_b$ is less than one second at $T_M$, and it is within several tens of minutes above $-50\,°C$. These dislocations can glide on non-basal planes until they turn to the dissociated state during deformation, finally resulting in sessile extended dislocations of various widths approaching to the equilibrium value $w_e$. In contrast, for perfect dislocations with Burgers vector <**a**>, $t_d$ is less than one second above $-50\,°C$, resulting in glissile extended dislocations with the equilibrium width $w_e$ on the basal plane. This width is sensitive to the shear stress $\tau$ exerted normal to the dislocation line, leading to extension of the intervening stacking fault across the entire crystal grain under commonly accessible stresses. Also, due to the widely dissociated state, dislocations <**a**> cannot cross-slip to non-basal planes. Such behavior of extended dislocations in ice are notable when compared to those of other materials.

**Keywords:** ice; extended dislocation; partial dislocation; stacking fault; plasticity

## 1. Introduction

Dislocations in hexagonal ice $I_h$ are known to widely extend on the basal plane, strongly restricting their glide and climb motion such that they lie only along this plane [1–4]. This restriction leads to the anisotropic nature of ice plasticity and also helps to explain the formation and annihilation of cubic ice $I_c$ (or stacking-disordered ice). The extended dislocations are generated through dissociation of a perfect dislocation into two partial dislocations bounding a stacking fault. As stacking fault energies in ice are very small compared to other materials, extended dislocations in ice have very large extended widths.

Overall, dislocation motion involves the undissociated state, dissociation, and migration of the partials. Given the large extended widths of dislocations in ice, the timescale may be long at low temperatures, leading to a strong temperature-dependence of ice plasticity. But how long does it take for a perfect dislocation to completely dissociate to its equilibrium separation? The purpose here is to provide a first calculation of this time for various perfect dislocations and temperatures. Before making this calculation, we first briefly summarize some relevant fundamental knowledge on dislocations in ice $I_h$, starting from the unit cell [1–6].

### 1.1. Burgers Vectors of Dislocations in Ice

Figure 1a shows the hexagonal unit cell of ice $I_h$, consisting of a tetrahedral arrangement of water molecules with basis vectors $\mathbf{a}_1$, $\mathbf{a}_2$, and $\mathbf{c}$. This crystal structure can be viewed as a layer structure with a stacking sequence *ABAB* of hexagonal ice $I_h$ as shown in (a). If the top bilayer *A* shifts to the bilayer *C* to occupy the empty site C-C′, the stacking sequence turns into *ABC* of cubic ice $I_c$. As the tetrahedral arrangement of water molecules is preserved in the cubic stacking sequence, ice $I_c$ is considered to be a metastable or transient state of ice $I_h$ [1,4–6]. Therefore, such a metastable (or transient) state must be taken into consideration when discussing Burgers vectors of dislocations in ice.

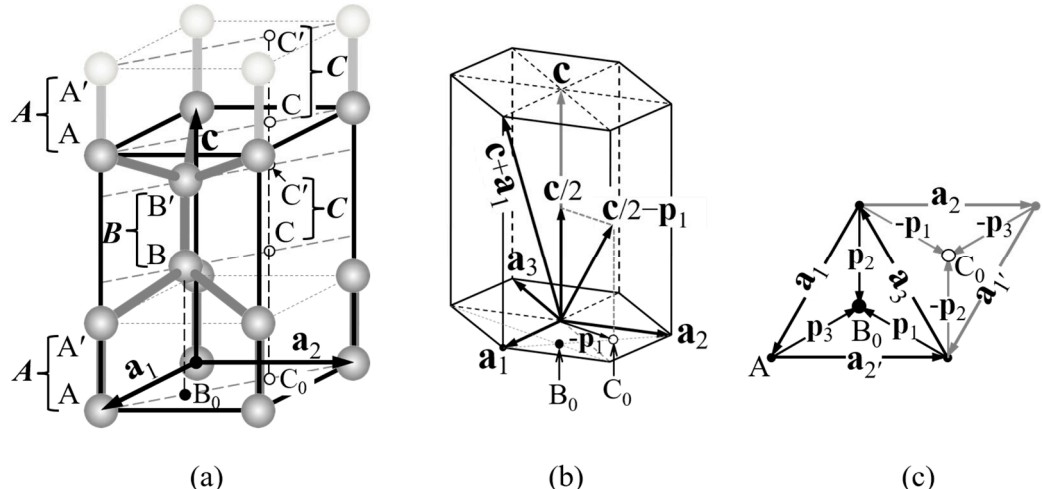

(a)　　　　　　　　　　　　　　(b)　　　　　　　　　　　　　　(c)

**Figure 1.** Burgers vectors in ice $I_h$. (**a**) Oblique view of the ball-and-stick model [7] for the ice $I_h$ crystal structure, with the basis vectors $\mathbf{a}_1$, $\mathbf{a}_2$, and $\mathbf{c}$ forming a hexagonal unit cell (the lattice parameters *a* and *c*). Balls (oxygen atoms) are connected by sticks (hydrogen bonds with one hydrogen per bond according to the ice rules [1,5,6]) to form the tetrahedral arrangement of water molecules. The sites $B_0$ and $C_0$ are the projections of the site B (and B′) and the empty site C (and C′) onto the basal plane on which $\mathbf{a}_1$ and $\mathbf{a}_2$ lie. Bimolecular layers A-A′, B-B′, and C-C′ are designated bilayers *A*, *B*, and *C*, respectively. (**b**) Burgers vectors <**c**>, <**a**>, and <**c** + **a**> for perfect dislocations, and <**p**>, <**c**/2>, and <**c**/2 + **p**> for partial dislocations. (**c**) Relations between the Burgers vectors <**a**> and <**p**> lying on the basal plane: $\mathbf{a}_i = \mathbf{p}_j - \mathbf{p}_k$ (*i, j, k* = 1, 2, 3 cyclic), with their magnitudes $|\mathbf{a}_i| = a$ and $|\mathbf{p}_i| = b_p = a/\sqrt{3}$.

Consider now the dislocations that can exist in ice $I_h$. In general, a dislocation has a Burgers vector **b** ($|\mathbf{b}| = b$) with a self-energy proportional to $b^2$. Thus, a smaller Burgers vector length *b* is energetically favored, resulting in the most common Burgers vector **b** in ice $I_h$ being equal to the basis vectors <**c**>, <**a**>, and their linear combination <**c** + **a**> as shown in Figure 1b. As these Burgers vectors are lattice translation vectors, the crystal structure translated by them are identical with the original structure, and therefore dislocations with these Burgers vectors are called perfect dislocations. Here, the vector bracketed by < > represents all equivalent vectors; for example, <**a**> and <**p**> represents $\mathbf{a}_i$ and $\mathbf{p}_i$ for *i* = 1, 2, and 3. In addition, the $\mathbf{a}_3$ axis is added in this figure for convenience of the four-index notation; i.e., <**a**> = $\frac{1}{3}\langle11\bar{2}0\rangle$ and <**p**> = $\frac{1}{3}\langle10\bar{1}0\rangle$.

On the other hand, dislocations with Burgers vectors that result in the cubic stacking sequence can also exist in ice $I_h$, and are called partial dislocations. Such Burgers vectors include horizontal translation vectors $\pm\mathbf{p}_i$ (*i* = 1, 2, 3) by which the atoms A-A′ or B-B′ move to the empty sites C-C′ in the same bilayer, and the oblique translation vectors $\mathbf{c}/2 \pm \mathbf{p}_i$ by which A-A′ or B-B′ move to C-C′ in the adjacent bilayer as shown in Figure 1b,c. In addition, translation by the vector $\mathbf{c}/2$ that can be made by inserting the bilayer *C* between bilayers *A* and *B* also results in the cubic stacking sequence. Thus, Burgers vectors for partial dislocations in ice are <**p**>, <**c**/2>, and <**c**/2 + **p**>.

### 1.2. Extended Dislocations in Ice

Partial dislocations are particularly important for ice plasticity. For example, a perfect dislocation $\mathbf{a}_3$ can dissociate into two partial dislocations of Burgers vectors $\mathbf{p}_1$ and $-\mathbf{p}_2$ to reduce the total self-energy ($a^2 > 2(b_\mathrm{p})^2$) according the relation $\mathbf{a}_3 = \mathbf{p}_1 - \mathbf{p}_2$ shown in Figure 1c. Figure 2 shows such dissociated states. Between the two partial dislocations, called Shockley partials, lies a plane defect called a stacking fault, in this case called a Shockley-type stacking fault and shown shaded in the bottom figure. The stacking fault can be viewed as a result of the imperfect shear displacement $\mathbf{p}_i$, which turns bilayer *A* into bilayer *C* as described in the preceding section. A side view of this fault shows how it results in a disrupted stacking sequence *ABC* as shown in the top figure, which is also the stacking sequence of cubic ice $I_\mathrm{c}$. The combination of partial dislocations and stacking fault is called an extended dislocation. Such an extended dislocation is glissile only on the basal plane because the Burgers vectors of two partial dislocations and the associated stacking fault lie on the basal plane, resulting in the strong restriction on its glide motion along the basal plane.

Consider a larger view of the molecular arrangements in and around the Shockley-type stacking fault. The top of Figure 2 shows the molecular positions in the fault as filled circles. The left and right edges are the Shockley partials projected onto the second prismatic plane $\left(11\bar{2}0\right)$. The bilayers marked *C* and *B* show a cubic stacking sequence between the partial dislocations, each of which is sandwiched by different bilayers (i.e., *ACB* and *CBA*, top to bottom). In contrast, adjacent hexagonal bilayers are sandwiched by the same bilayers (e.g., *ABA*). Thus, a Shockley-type stacking fault is associated with two cubic bilayers.

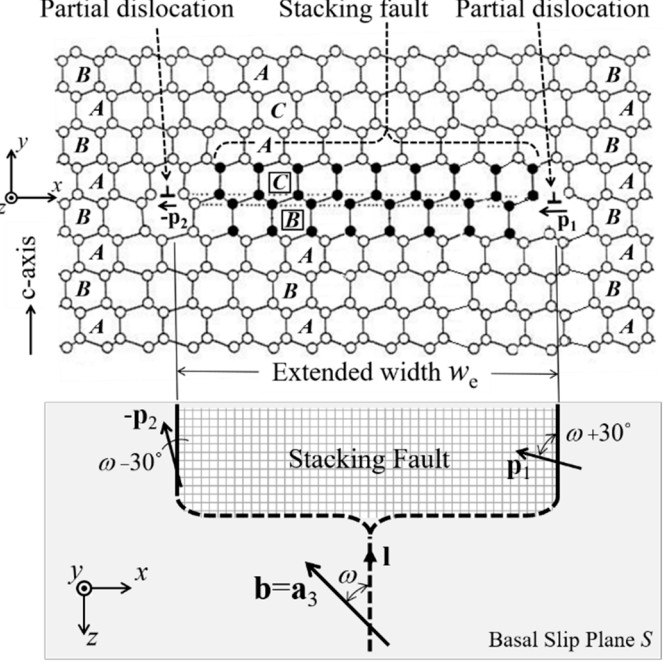

**Figure 2.** An extended dislocation glissile on the basal plane. (Top) Molecular arrangement projected onto $\left(11\bar{2}0\right)$. The Shockley-type stacking fault is composed of two cubic-stacked bilayers *B* and *C* as indicated by filled circles between the two partials. (Bottom) An arrangement of Shockley partials $\mathbf{p}_1$ and $-\mathbf{p}_2$ (see Figure 1c) dissociated from a perfect dislocation with Burgers vector $\mathbf{b} = \mathbf{a}_3$, with the angle $\omega$ between $\mathbf{b}$ and the dislocation line $\mathbf{l}$ [4,8].

Perfect dislocations with the Burgers vectors <**c**> and <**c** + **a**> can also dissociate into two partial dislocations bounding a stacking fault. Their dissociations are $\mathbf{c} \rightarrow \mathbf{c}/2 + \mathbf{c}/2$, $\mathbf{c} \rightarrow (\mathbf{c}/2 + \mathbf{p}_i) + (\mathbf{c}/2 - \mathbf{p}_i)$, and $\mathbf{c} + \mathbf{a}_i \rightarrow (\mathbf{c}/2 + \mathbf{p}_j) + (\mathbf{c}/2 + \mathbf{p}_k)$ as one can see using Figure 1b,c [8]. However, unlike the Shockley type above, the Burgers vectors of the partials in the present cases have a normal component to the

stacking-fault plane. The first set of partials with Burgers vectors **c**/2, being normal to the fault plane, are called Frank partials; the other ones are inclined to the fault plane and are thus examples of Frank–Shockley partials.

Consider first a simple dissociation from a perfect dislocation <**c**> to two Frank partials <**c**/2>. Figure 3a shows insertion of an extra bilayer *C*, just one side shown with one edge the Frank partial <**c**/2>. Between the two partials lies the Frank-type stacking fault associated with three cubic bilayers $\boxed{B}$, $\boxed{C}$, and $\boxed{A}$.

Similarly, a perfect dislocation <**c** + **a**> dissociates into two Frank–Shockley partials <**c**/2 + **p**>, one side of which is shown in Figure 3b. Between the two partials lies a Frank–Shockley type stacking fault, but unlike the previous case, this fault is associated with only one cubic bilayer *A*. This case can be constructed by introducing a pair of Shockley partials with $\pm\mathbf{p}_i$ into the case shown in (a). That is, the Frank type with three cubic bilayers changes to a Frank–Shockley type with one cubic bilayer. Although the original perfect dislocations <**c**> and <**c** + **a**> are glissile on prismatic and pyramidal planes respectively, the resulting extended dislocations are sessile because, unlike the Shockley partial, their Burgers vectors have normal components to the fault plane (basal plane).

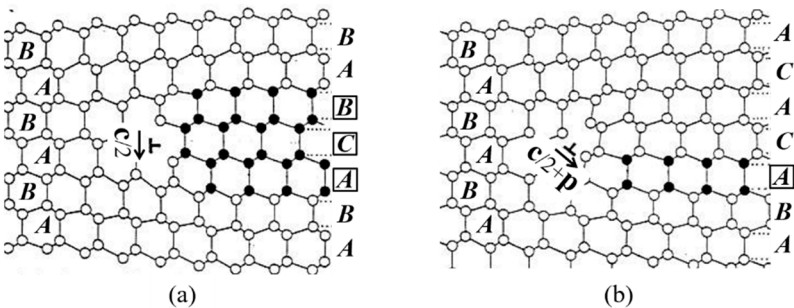

(a)                                      (b)

**Figure 3.** Sessile partial dislocations. (**a**) Frank partial dislocation with Burgers vector <**c**/2> associated with three bilayers of cubic sequence $\boxed{B}$, $\boxed{C}$, and $\boxed{A}$. (**b**) Frank–Shockley partial dislocation with Burgers vector <**c**/2 + **p**> associated with mono-bilayer of cubic sequence $\boxed{A}$ [4,8].

### 1.3. Basal Slip System in Ice

The basal slip system in ice has two types of slip planes $S_0$ and $S$ (or $S'$) with different interplanar spacings as shown in Figure 4a. In general, dislocations on $S$ or $S'$ are termed the glide set, whereas those on $S_0$ are called the shuffle set [1,4,9–13]. According to the Peierls–Nabarro model [9], a wider interplanar spacing results in a smaller Peierls stress (a smaller lattice frictional stress), and therefore the shuffle set, which has a wider interplanar spacing than $S$ and $S'$, should dominate over the glide set during basal shear deformation. However, the extended dislocation shown in Figure 2 can move only on the glide set slip planes $S$ or $S'$ because the stacking fault bounded by Shockley partials can exist and move only on $S$ or $S'$, not on $S_0$. It will be discussed later (Section 2.4) which of the glide or shuffle set is dominant in the basal slip system in ice.

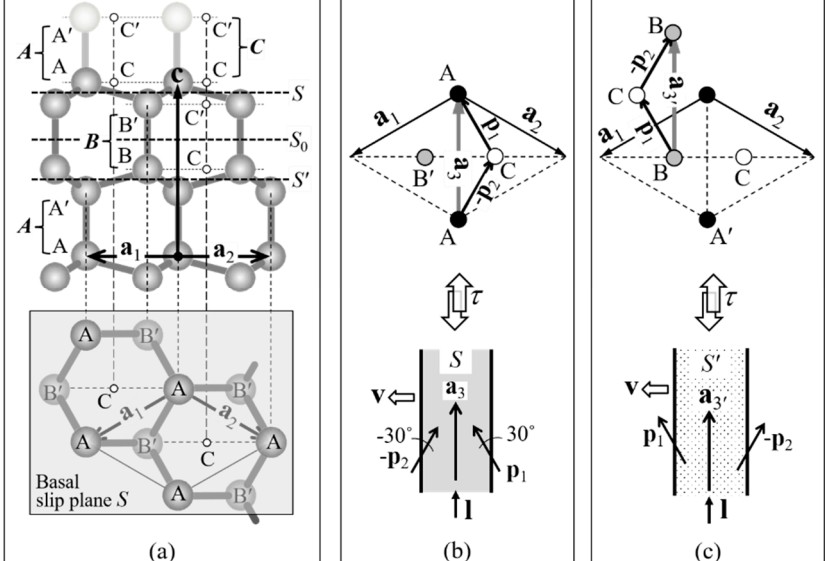

**Figure 4.** Basal slip system. (**a**) (Top) Glide set *S* (and *S'*) and shuffle set $S_0$ for the basal slip system in $I_h$. (Bottom) Top view showing the relative position of the atoms just above and below the slip plane *S*. (**b**) An extended dislocation on the slip plane *S*, with the Shockley partials $\mathbf{p}_1$ and $-\mathbf{p}_2$ dissociated from a right-handed screw dislocation $\mathbf{a}_3$. Glide motion in the direction **v** results in the two-step displacement of the top half above *S* by $-\mathbf{p}_2$ followed by $\mathbf{p}_1$ relative to the bottom half. (**c**) The same as (b) but with the exchanged arrangement of the partials on the slip plane *S'*.

The extended dislocations shown in Figure 4b,c can move only on the glide set slip planes *S* and *S'*, respectively. In contrast, a basal perfect dislocation <**a**> moves only on the shuffle set slip plane $S_0$. Extended dislocations in the shuffle set have been theoretically proposed [9], but these have not been verified. Therefore, we assume on dislocations with Burgers vector <**a**> that all dissociated dislocations are of the glide set, and all undissociated dislocations are of the shuffle set. According to extensive studies of dislocations in other semiconductor materials with tetrahedral bonding, extended glide-set dislocations are active in deformation at high temperatures and low stresses, whereas perfect shuffle set dislocations are the more active type at low temperatures and high stresses [10–13].

Extended dislocations on *S* differ from those on *S'*, with different arrangements of Shockley partials. For example, Figure 4b shows the extended dislocation moving on *S* in the direction **v** under an applied shear stress $\tau$. The top drawing shows shear displacement $-\mathbf{p}_2$ being followed by $\mathbf{p}_1$, moving the top half above the slip plane *S* over the bottom half via a two-step displacement. By this two-step process, the bilayer A-A' moves by $\mathbf{a}_3$ via the empty site C-C' (above $C_0$) shown in (a) and (b), resulting in the stacking sequence *ABC* of $I_c$ between the two Shockley partials. Although the *ABAB* sequence of $I_h$ has a lower energy, the energy difference between the two sequences is very small because the tetrahedral bonding is preserved by the shear displacement. In contrast, if the order of the two-step displacement is reversed, the bilayer A-A' has to move via the site B-B', resulting in the stacking sequence *ABB*, which breaks the bonds between *B* and *B*, involving much greater energy. Therefore, the two-step displacement should take the path via the site C-C', resulting in the arrangement of Shockley partials shown at bottom in (b). The situation is reversed on *S'*, with the two-step displacement made by $\mathbf{p}_1$ followed by $-\mathbf{p}_2$ being energetically favored as shown in (c). These cases involved a right-handed screw dislocation for the perfect dislocation $\mathbf{a}_3$; that is, the line sense vector **l** is aligned with the Burgers vector $\mathbf{b} = \mathbf{a}_3$.

In general, the glide motion of a dislocation with Burgers vector **b** on the slip plane *S* in the direction **v** displaces the top half above *S* by the Burgers vector **b** with respect to the bottom half when the vector product $\mathbf{n} = \mathbf{l} \times \mathbf{v}$ is directed from the bottom toward the top (according to axiom 1.3 of [9]). This is the case in Figure 4b,c. On the other hand, when the direction of shear stress is

reversed, the extended dislocation moves in the opposite direction, $-\mathbf{v}$ in (b), $\mathbf{n}$ reverses, and therefore the bottom half moves by $\mathbf{p}_1$ followed by $-\mathbf{p}_2$ relative to the top half, resulting in the same two-step displacement shown in (b).

*1.4. Equilibrium Widths of Extended Dislocations in Ice*

Consider now the stacking-fault energies. By measuring the shrinkage rates of dislocation loops in $I_h$, the stacking-fault energy $\gamma_{f1}$ for the Frank–Shockley type shown in Figure 3b was determined to be 0.31 mJ/m$^2$ at −20 °C [14]. No measurements were done for the other two stacking-fault types. Assuming that their energy is proportional to the number of cubic bilayers with the same prefactor, then the stacking-fault energy for the Shockley type is $\gamma_{f2} = 2\gamma_{f1} = 0.62$ mJ/m$^2$ and the Frank type is $\gamma_{f3} = 3\gamma_{f1} = 0.93$ mJ/m$^2$. As the Shockley type stacking-fault plays a key role in plastic deformation and its fault vector equals <**p**>, we use the notation $\gamma_p$ instead of $\gamma_{f2}$ in the following sections. In support of the measurements, simulations with the mW water model [15,16] found a $\gamma_{f1}$ value of $0.30 \pm 0.05$ mJ/m$^2$. Considering an error of about 10% included in the above experimentally determined value [14], these two values show fairly good agreement. Thus, without further knowledge of their temperature dependences, we assume the above values for lower temperatures as well.

These energies are useful for calculating the width of the extended dislocation in equilibrium $w_e$. This is done by equating the repulsive force between the two partial dislocations with the shrinkage force due to the stacking-fault energy $\gamma_{SF}$ representing one of the above values. The resulting width depends on the Burgers vectors $\mathbf{b}_1$ and $\mathbf{b}_2$ of partial dislocations with line vectors $\mathbf{l}_1$ and $\mathbf{l}_2$ as [9]

$$w_e = \frac{\mu}{2\pi\gamma_{SF}}\left[(\mathbf{b}_1\bullet\mathbf{l}_1)(\mathbf{b}_2\bullet\mathbf{l}_2) + \frac{(\mathbf{b}_1\times\mathbf{l}_1)\bullet(\mathbf{b}_2\times\mathbf{l}_2)}{1-\nu}\right], \tag{1}$$

where $\mu$ and $\nu$ are the shear modulus and Poisson's ratio of ice. For the case of Shockley partials dissociated from a perfect dislocation with Burgers vector $\mathbf{b}$ = <**a**> shown in Figure 2,

$$w_e = \frac{\mu b_p^2}{8\pi\gamma_p}\frac{2-\nu}{1-\nu}\left(1 - \frac{2\nu\cos(2\omega)}{2-\nu}\right) \tag{2}$$

where $b_p$ is the Burgers vector length of Shockley partials, and $\omega$ is the angle between Burgers vector $\mathbf{b}$ and the line vector $\mathbf{l}$ of the (undissociated) perfect dislocation. For the Frank type shown in Figure 3a,

$$w_e = \frac{\mu}{8\pi\gamma_{f3}}\frac{c^2}{1-\nu} \tag{3}$$

For the Frank–Shockley type,

$$w_e = \frac{\mu}{8\pi\gamma_{f1}}\frac{1}{1-\nu}\left[c^2 - 4b_p^2\left(1 - \nu\cos^2\omega_p\right)\right], \tag{4}$$

for the case when dissociated from a perfect dislocation <**c**> as $\mathbf{c} \to (\mathbf{c}/2 + \mathbf{p}_i) + (\mathbf{c}/2 - \mathbf{p}_i)$ with an angle $\omega_p$ between Burgers vector and the line vector of the partials, and

$$w_e = \frac{\mu}{8\pi\gamma_{f1}}\frac{1}{1-\nu}\left[c^2 + b_p^2(2 - \nu - 2\nu\cos(2\omega))\right], \tag{5}$$

for the dissociation from a perfect dislocation <**c** + **a**> as $\mathbf{c} + \mathbf{a}_i \to (\mathbf{c}/2 + \mathbf{p}_j) + (\mathbf{c}/2 - \mathbf{p}_k)$ (see Figure 1c).

In Equations (2) to (5), we use the temperature-range-averaged constant values for the shear modulus $\mu$ and lattice constants $a$ and $c$ averaged over the values at temperatures 160, 190, 220, and 250 K. For simplicity, we adopt these constant values throughout this paper; i.e., $a$ = 0.451 nm, $c$ = 0.734 nm, $\mu$ = 3.78 (GN/m$^2$), and $\nu$ = 0.325 ($\mu$ changes by ~10% over this temperature range) [17,18], resulting in the magnitude of Burgers vectors $b_p$ = 0.260 nm for Shockley partials, $c/2$ = 0.367 nm for

Frank partials, and $\{(c/2)^2 + (b_p)^2\}^{1/2} = 0.450$ nm for Frank–Shockley partials. Extended widths $w_e$ calculated for different dissociation reactions are shown in Table 1.

**Table 1.** Extended dislocations in ice.

| Plane | Perfect Dislocation | Dissociation Reaction | Extended Width [*1] $w_e$ |
|:---:|:---:|:---:|:---:|
| (0001) | <**a**> | $\mathbf{p}_j + (-\mathbf{p}_k)$ | 25 nm (<**a**>: screw)<br>49 nm (<**a**>: 60°)<br>57 nm (<**a**>: edge) |
| (0001) | <**c**> | $\mathbf{c}/2 + \mathbf{c}/2$ | 129 nm |
| | | $(\mathbf{c}/2 + \mathbf{p}_i) + (\mathbf{c}/2 - \mathbf{p}_i)$ | 256 nm (<**p**>: screw)<br>193 nm (<**p**>: edge) |
| (0001) | <**c + a**> | $(\mathbf{c}/2 + \mathbf{p}_j) + (\mathbf{c}/2 - \mathbf{p}_k)$ | 437 nm (<**a**>: screw)<br>501 nm (<**a**>: edge) |

[*1] The extended widths calculated by Equations (2) to (5) for the shear modulus $\mu$ and lattice constants $a$ and $c$ averaged over the values at temperatures 160, 190, 220, and 250 K.

## 2. Formation of a Glissile Extended Dislocation

### 2.1. Driving Force for Dissociation of Perfect Dislocation <a>

The equilibrium extended widths $w_e$ in Table 1 are all much larger than the Burgers vector length. This suggests that once a new perfect dislocation is introduced into a crystal, a considerable time may be needed to attain the equilibrium separation. But how long is this time? Here, we call this time the dissociation-completing time $t_d$. We start by considering a perfect dislocation with Burgers vector <a>.

First consider the force. Figure 2 shows two Shockley partials exerting a repulsive force (per unit length of dislocation) on each other as expressed by [9]

$$
\begin{aligned}
f_{edge} &= \frac{\mu b_1 b_2}{2\pi(1-\nu)} \frac{x(x^2-y^2)}{(x^2+y^2)^2} \\
&= \frac{\mu b_p^2 \sin(\omega-30°)\sin(\omega+30°)}{2\pi(1-\nu)r} \\
&= \frac{\mu b_p^2 \{1 - 2\cos 2\omega\}}{8\pi(1-\nu)r}
\end{aligned}
\tag{6}
$$

where $\mu$ is the shear modulus, $\nu$ the Poisson's ratio, $b_p$ the Burgers vector length of the Shockley partial, and $\omega$ is the angle between the Burgers vector **b** (i.e., $\mathbf{a}_3$) and the dislocation line **l** of the perfect dislocation as shown in Figure 2. The above case involves an edge component with the extended dislocation lying on the $xz$-plane, and the partials separated by a distance $r$ ($<w_e$) running parallel to the $z$-axis as shown in the figure. For a screw component, the force is instead

$$
f_{screw} = \frac{\mu b_p^2 \{1 + 2\cos 2\omega\}}{8\pi r}.
\tag{7}
$$

In addition, an attractive force per unit length of dislocation arises that is equal to the stacking fault energy of Shockley type $\gamma_p$ per area. Here, $\gamma_p = \gamma_{f2}$ as described in the preceding subsection. Summing these together, the net driving force exerting on each partial is given by

$$
f(r) = \frac{\mu b_p^2 \{2 - \nu(1 + 2\cos 2\omega)\}}{8\pi(1-\nu)r} - \gamma_p.
\tag{8}
$$

For simplicity, we adopt constant values for the shear modulus $\mu$ and lattice constants $a$ and $c$ (i.e., Burgers vector length $b$) averaged over the calculated values at temperatures 160, 190, 220, and 250 K ($\mu$ changes by ~10% over this temperature range) [17,18]. With $\mu = 3.78$ (GN/m²), $\nu = 0.325$, $b_p = 0.26$ (nm), $\gamma_p = 0.62$ (mJ/m²), and setting $f(r) = 0$, we have $r = w_e = 25$ (nm) for a screw perfect dislocation ($\omega = 0°$).

### 2.2. Dissociation-Completing Time

By definition, the dissociation-completing time $t_d$ is

$$t_d = \int_{R_0}^{w_e - R_0} \frac{1}{V(r)} dr \tag{9}$$

where $V(r)$ is the dislocation speed and $R_0$ ($\ll w_e$) is introduced to avoid divergence in the integration (we set $R_0$ in Equation (9) to 0.1 nm.) From [19,20], $V(r)$ and the dislocation mobility $M$ obey

$$\begin{aligned} V(r) &= M(T)f(r) \\ M(T) &= M_0 \exp\left(-\frac{E}{k_B T}\right) \end{aligned} \tag{10}$$

where $T$ is the temperature, $E$ the activation energy of glide motion, and $k_B$ is Boltzmann's constant. In this way, $t_d$ can be calculated as a function of temperature for a fixed angle $\omega$. As the mobility depends on the mechanism of the glide motion, we consider a range of parameters with $M_0 = 3 \times 10^3$ (m$^2$/Ns) and $E = 0.756$ (eV) for the lower bound, and $M_0 = 14.46$ (m$^2$/Ns) and $E = 0.624$ (eV) for the upper bound [1]. In this calculation, the $M_0$ value for a Shockley partial dislocation is assumed to be twice that for a perfect dislocation because it dissociates into two partial dislocations with the same mobility according to [21].

The resulting $t_d$ values increase rapidly with decreasing temperature as shown in Figure 5. The thick solid line 1 and the thick dashed line 2 show $t_d$ for a perfect screw dislocation (i.e., $\omega = 0°$) calculated by using the above mobility data of the upper and lower bound, respectively. Due to the exponential temperature dependence of the mobility, $t_d$ is very large at low temperatures. For example, below $-150$ °C (123 K), Figure 4 shows that $t_d$ exceeds $10^3$ years for both cases, but above $-50$ °C (223 K) it is less than one second.

### 2.3. Dissociation-Beginning Time

As the dissociation-completing time $t_d$ is very large at low temperatures as described above, a new problem arises. How much time does it take to transform a perfect dislocation (in an undissociated state) into a dissociated state? To determine whether dislocations would likely be observed in an undissociated or dissociated state, we define the dissociation-beginning time $t_b$ as the time required for making a dissociation distance large enough to be clearly distinguished from the undissociated state. Concerning the core diameters of Shockley partials in ice, reconstructed cores of 30°- and 90°-Shockley partials given by the DFT calculations [22] have widths of about 1.7$a$ and 2.3$a$ on the basal plane, respectively. When a monolayer of regular hexagonal ring with a thickness of about $a$ is inserted between the two cores, these two partials can be viewed as in a well separated state. Therefore, the distance is fixed to be three times the lattice constant $a$, and thus any separation larger than this can be viewed as a dissociated state. We further assume here that the time to nucleate, or start the dissociation is negligible in comparison. Then, to calculate $t_b$, we replace $w_e$ in Equation (9) by 3$a$. The result is given by the thin solid line 1′ and the thin dashed line 2′ in Figure 5. For example, the dissociation begins within 1 day at around 150 K, whereas it takes 10 days to a few years to complete the dissociation. The numerical data on all four curves at typical temperatures are shown in Table 2.

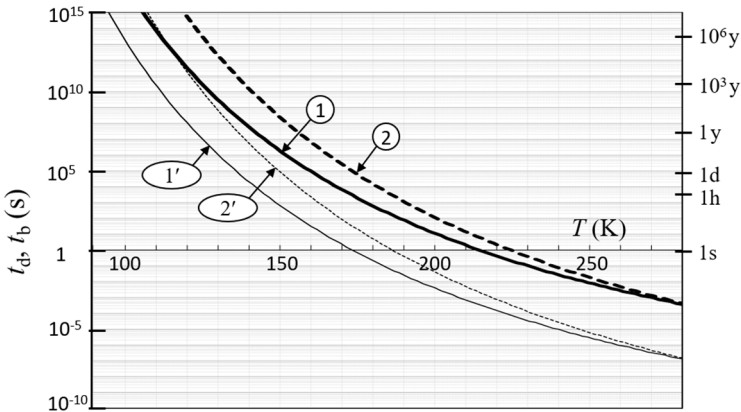

**Figure 5.** Calculated dissociation-beginning times (thin curves) and dissociation-completing times (thick curves) for a range of temperatures. The calculations assumed a basal extended dislocation with Burgers vector <**a**> with the larger-mobility value (solid curves) and smaller-mobility value (dashed curves) for Shockley partials. The dissociation-beginning time $t_b$ (curves 1' and 2') is the time needed to separate the two Shockley partials to a distance of $3a$. The dissociation-complete time $t_d$ (curves 1 and 2) is the time needed to separate the two Shockley partials to a distance of the equilibrium separation $w_e$. Numerical data of $t_d$ and $t_b$ are in Table 2.

The above times are useful for estimating the expected widths of the extended dislocations. For example, on a one-second timescale, perfect dislocations with Burgers vector <**a**> should stay totally undissociated below about −150 °C, but completely dissociate above about −50 °C. At this timescale, the dissociation-beginning temperature lies roughly within −100 and −87 °C. As a result, the glissile extended dislocations should have a wide range of extended widths up to their equilibrium value $w_e$ in the temperature range of −150 to −50 °C, although these temperatures will be affected by the timescale of other physical processes available to the dislocation. For example, if a perfect screw dislocation with the Burgers vector $\mathbf{a}_2$ shown in Figure 1b is introduced at low temperatures, it can cross slip to the prismatic $(10\bar{1}0)$, pyramidal $(10\bar{1}1)$, and $(10\bar{1}\bar{1})$ planes because of its undissociated form, whereas such cross slips are not allowed in its dissociated form due to being widely extended on the basal plane. Thus, the behavior of basal dislocations <**a**> under deformation could be different at low temperatures.

**Table 2.** Calculated dissociation-completing time $t_d$ and dissociation-beginning time $t_b$ for basal extended dislocations with Burgers vector <**a**>.

| Case | | Temperature | | | | |
|------|---|---|---|---|---|---|
| | | 253 K (−20 °C) | 223 K (−50 °C) | 193 K (−80 °C) | 173 K (−100 °C) | 123 K (−150 °C) |
| $t_d$ [*1] | 1 | $5.8 \times 10^{-3}$ s (5.8 ms) | $2.7 \times 10^{-1}$ s (0.27 s) | $4.2 \times 10$ s (42 s) | $3.2 \times 10^3$ s (0.89 h) | $7.9 \times 10^{10}$ s (2.5 Ky) |
| | 2 | $1.2 \times 10^{-2}$ s (12 ms) | $9.7 \times 10^{-1}$ s (0.97 s) | $5.7 \times 10^2$ s (570 s) | $1.1 \times 10^4$ s (4 h) | $9.8 \times 10^{13}$ s (3.1 My) |
| $t_b$ [*2] | 1' | $2.0 \times 10^{-6}$ s (2.0 μs) | $2.0 \times 10^{-4}$ s (0.2 ms) | $1.5 \times 10^{-2}$ s (15 ms) | $1.1$ s (1.1 s) | $2.7 \times 10^7$ s (0.86 y) |
| | 2' | $4.1 \times 10^{-6}$ s (4.1 μs) | $9.2 \times 10^{-4}$ s (0.92 ms) | $0.20$ s (0.20 s) | $3.8 \times 10^1$ s (38 s) | $3.4 \times 10^{10}$ s (1.1 Ky) |

[*1] Dissociation-completing time: time needed to form an extended dislocation (Burgers vector **a**) with an equilibrium separation $w_e$ between two Shockley partials. Result 1 is based on the upper mobility bound, 2 on the lower.
[*2] Dissociation-beginning time: time needed to separate the two Shockley partials to a distance of three times the Burgers vector length $a$. Result 1' is based on the upper mobility bound, 2' on the lower.

### 2.4. Shuffle-Glide Transformation and Nucleation of Shockley Partials in Ice

The dissociation process can also involve a transformation of the glide plane from the shuffle set ($S_0$) to the glide set ($S$ or $S'$). This transformation should occur when a perfect dislocation dissociates into Shockley partials, a transformation that requires not only rearrangement of molecules in the dislocation core, but also mass transport due to climb motion by the distance $c/2$. This mass transport is very slow at low temperatures. Thus, at sufficiently low temperatures, the dissociation-beginning temperature described above will underestimate the observed times because this climb motion from $S_0$ to $S$ will begin to limit the dissociation process. Due to the difficulty of making this transformation, a shuffle-set perfect dislocation can stay undissociated at low temperatures even though the transformation into the glide-set extended dislocation is energetically preferred. Hence, we can refer to this process as the 'nucleation' of the dissociated state, which may involve formation of a double jogs. Consequently, such an undissociated basal dislocation can move through $S_0$ (i.e., the shuffle set) at low temperatures, resulting in a lower Peierls stress for the glide motion at low temperatures.

However, there are ways that widely extended dislocations can be nucleated during deformation at low temperatures. Consider the case in which deformation nucleates a basal dislocation at a free surface or a grain boundary. Nucleation at such a boundary may favor Shockley partials over perfect dislocations because they have a shorter Burgers vector and hence should have a smaller threshold energy. Then, one Shockley partial moves into the crystal, with the second one nucleated when the width of the stacking fault becomes larger than $w_e$. The end result is an extended dislocation with a separation approximately equal to $w_e$, even at low temperatures. This analysis indicates that low-temperature ice can have a variety of basal dislocations, such as ones in the glide set with different extended widths and ones in the shuffle set without dissociation. In contrast, at temperatures above about −50 °C and over laboratory timescales, ice may typically have only the extended glide set basal-dislocation.

### 2.5. Extended Width Changing under a Shear Stress

In considering other relevant processes, we should also examine opposite directions in the glide motion of the Shockley partials of an extended dislocation under an applied shear stress. For example, consider the extended screw dislocation shown in Figure 6a. Here, a shear stress $\tau$ ($= \sigma_{xy}$) is applied on the plane of the stacking fault, normal to the dislocation line. Although the undissociated screw dislocation is subjected to no force under the shear stress, the two 30°-Shockley partials with edge components of their Burgers vector $\mathbf{b}_e$ and $-\mathbf{b}_e$ respond by moving in opposite directions. As shown in Figure 6b, their motion extends the separation on $S$, but shrinks the separation on $S'$. As expected for a shear stress normal to a screw dislocation line, the total force is zero. But, the force $\tau b_e$ on each partial adds to the force from the stacking fault. Thus, to calculate a new equilibrium separation $r_e$, we again use Equation (2), but replace $\gamma_{SF}$ with $\gamma_p \pm \tau b_p/2$, where '+' is for the $S'$ plane and '−' for $S$ (here, we use $b_e = b_p \sin 30° = b_p/2$ with the Burgers vector length $b_p$ of the Shockley partial). This change of Equation (2) with $\omega = 0°$ gives a new equilibrium width $r_e$ of

$$r_e = \frac{\mu p^2 (2 - 3\nu)}{8\pi(1-\nu)\left(\gamma_p \pm \frac{\tau b_p}{2}\right)}. \tag{11}$$

Figure 6b shows both cases. As shown in the figure, $\Delta r \equiv |r_e - r|$ for the $S'$ plane and $\Delta r'$ for the $S$ plane. If $\tau b_p/2 << \gamma_p$, no change is expected by the applied shear stress $\tau$; that is, $\Delta r$ and $\Delta r'$ are much less than $r$.

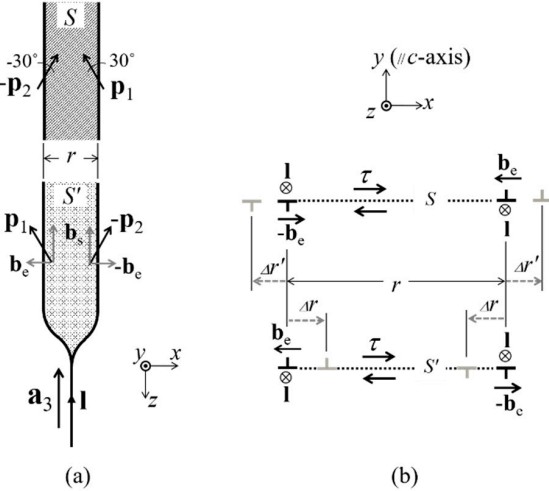

**Figure 6.** Separation changes of an extended screw dislocation under an applied shear stress. (**a**) Burgers vectors of Shockley partials on the basal planes $S'$ and $S$ (see Figure 4b,c). The $z$-axis is parallel to the line sense $\mathbf{l}$, whereas $y$ is normal to the basal plane. The vectors $\mathbf{b}_e$ and $\mathbf{b}_s$ are the edge and screw components of the Burgers vector $\mathbf{p}_1$ of the Shockley partial. (**b**) The arrangement of edge components, viewed parallel to the $z$-axis. The applied shear stress $\tau$ exerts a force $\tau b_e$ on the dislocations, causing the two partials to move apart by $2\Delta r'$ on $S'$, but to move closer by $2\Delta r$ on $S$.

However, with ice, the stress term can be significant. The estimated value of $2\gamma_p/b_p \approx 4.8$ MPa, which is less than the applied stresses in deformation experiments (e.g., [23]). When $\tau$ approaches $2\gamma_p/b_p$ (or 4.8 MPa), $r_e$ on $S$ becomes infinite. In this situation, the $S$-type stacking faults must spread over the entire crystal grain. However, when $\tau \geq 2\gamma_p/b_p$, then $r_e$ on $S'$ becomes less than half $w_e$. This characteristic behavior of extended dislocations in ice comes from its remarkably low stacking-fault energy compared to other materials. As a result, the value of $2\gamma_p/b_p$ in ice is much smaller than that of other materials (Table 3), and thus, unlike the situation in many other materials, commonly applied stresses can cause some stacking faults to greatly expand (or shrink) and affect ice's plasticity. In addition, if during deformation many basal screw dislocations pile-up against some obstacle such as a grain boundary, then an effective shear stress $\tau$ on the top dislocation is multiplied by the number of piled-up dislocations [9], and can be much larger than $2\gamma_p/b_p$, resulting in the transition to the shuffle set. They can cross slip to prismatic $\{10\bar{1}0\}$ or pyramidal $\{10\bar{1}1\}$ planes.

**Table 3.** Parameters involved with dissociation into two Shockley partials of a perfect screw dislocation in various materials.

| Material | Shear Modulus $\mu$ (GPa) [*1] | Poisson's Ratio $\nu$ [*1] | Burgers Vector $b_p$ (nm) [*2] | SF Energy $\gamma_p$ (mJ/m²) [*3] | Extended Width $w_e$ (nm) | $2\gamma_p/b_p$ (MPa) |
|---|---|---|---|---|---|---|
| Ice ( I$_h$ ) | 3.55 | 0.325 | 0.261 | 0.62 | 25 | 4.8 |
| Si (diamond) | 68.1 | 0.218 | 0.222 | 55 | 4.2 | 500 |
| Ge (diamond) | 56.4 | 0.200 | 0.231 | 60 | 3.5 | 520 |
| CdS (wurzite) | 18.5 | 0.378 | 0.239 | 8.7 | 6.7 | 73 |
| Cu (fcc) | 54.6 | 0.324 | 0.148 | 45 | 1.6 | 610 |
| Zn (hcp) | 43.4 | 0.249 | 0.154 | 140 | 0.5 | 1800 |

[*1] Elastic constants: Gammon et al. [18] for ice; Anderson et al. [10] for Si, Ge, Cu, and Zn; Tan et al. [24] for CdS.
[*2] Lattice constants: Röttger et al. [17] for ice, Cullity [25] for Si, Ge, Cu, and Zn; Tan et al. [24] for CdS. [*3] SF energy: Hondoh et al. [14] for ice; Holt and Yacobi [12] for Si, Ge, and CdS; Anderson et al. [10] for Cu and Zn.

In most of the other materials, this effect can be neglected. For example, as commonly applied stresses used in deformation experiments of Si at high temperatures are smaller than 100 MPa, the second term $\tau b_p/2$ can be neglected because $\tau \ll 2\gamma_p/b_p \approx 500$ (MPa) for Si (Table 3). Perhaps this experience with other materials has led to the effect also being neglected in ice, but in ice it can significantly affect ice plasticity.

## 3. Formation of a Sessile Extended Dislocation

Through X-ray diffraction topographic observations of faulted and unfaulted dislocation loops during various heat treatments [2,3,14,26,27] it has been established that self-interstitials play a dominant role in mass transport in ice over vacancies, with experimental determinations of equilibrium concentrations $C_e$ [28] and diffusion coefficients $D_{SI}$ [29,30] of self-interstitials as functions of temperature and pressure. It was also confirmed that the self-diffusion coefficients calculated by these data were approximately equal to those determined by the tracer diffusion experiments [31]. To calculate the dissociation times $t_b$ and $t_d$ for sessile extended dislocations, therefore, we consider diffusive migration of self-interstitials between two partial dislocations during dissociation process from a perfect dislocation <**c**> or <**c** + **a**>. Although Frank–Shockley partial dislocation loops with Burgers vector <**c**/2 + **p**> are easily formed by segregation of self-interstitials by cooling [14,27], here we focus on dissociation from a straight perfect dislocation with Burgers vector <**c** + **a**>, which plays an important role in non-basal deformation.

### 3.1. Driving Force for Dissociation of Perfect Dislocations <c> and <c + a>

Extended dislocations arising from perfect dislocations with Burgers vector <**c**> and <**c** + **a**> cannot move without mass transport because Burgers vectors of partial dislocations do not lie on the plane of stacking fault as shown in Figure 7a, resulting in the name 'sessile extended dislocation'. Therefore, their dissociation involves climb motion of partial dislocations bounding the stacking fault.

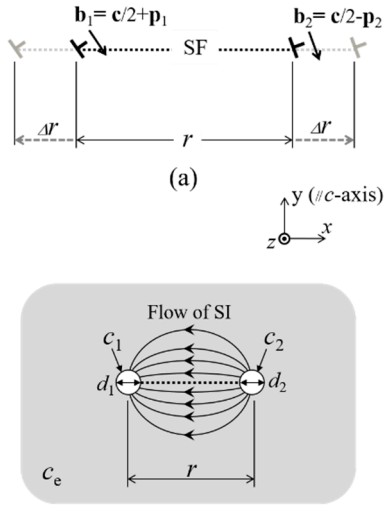

**Figure 7.** Dissociation of a sessile perfect dislocation. (**a**) Climb motion of Frank–Shockley partials **b**$_1$ (= **c**/2 + **p**$_1$) and **b**$_2$ (= **c**/2 − **p**$_2$), with **p**$_1$ − **p**$_2$ = **a**$_3$ (Figure 1c), in a basal plane that dissociated from a perfect dislocation <**c** + **a**>, with stacking fault (SF) lying in between. They move apart due to a mutually repulsive force. (**b**) A self-interstitial (SI) flow model for the dissociation process. The partials **b**$_1$ and **b**$_2$ are replaced by cylinders of diameters $d_1$ and $d_2$ embedded in infinite medium. SI flow occurs from cylinder 2 to 1, driven by a difference in SI concentrations $C_1$ ($<C_e$) and $C_2$ ($>C_e$) at the cylinder surfaces, where $C_e$ is the equilibrium concentration at temperature $T$.

Figure 7a shows climb motion of the Frank–Shockley partials due to a repulsive interaction force between the two partials during dissociation of an edge perfect dislocation with Burgers vector <**c** + **a**>. The Frank–Shockley type stacking fault lies on the basal plane as shown in Figure 3b, but the Burgers vector <**c**/2 + **p**> of the two partials bounding the stacking fault has a normal component to the basal plane as shown in Figure 7a. Therefore, climb motion of the partials is required for the dissociation. Our aim here is to calculate the time required to dissociate by climb.

Consider the dissociation process of a perfect dislocation with Burgers vector <**c** + **a**>. Figure 7a shows the two dislocations a distance $r$ apart. The two Frank–Shockley partials repel each other with a force (per unit length of dislocation) at distance $r$ given by

$$f(r) = \frac{\mu(2+v)p^2}{8\pi(1-v)r} + \frac{\mu c^2}{8\pi(1-v)r} - \gamma_{f1},$$ (12)

where $\mu$ is the shear modulus, $v$ the Poisson ratio, $\gamma_{f1}$ the stacking fault energy for the fault vector <**c**/2 + **p**>, and $p$ and $c$ the Burgers vector lengths of <**p**> and <**c**>. Here, the Burgers component <**a**> is normal to the dislocation line. Under $f(r)$, the two partials **b**$_1$ and **b**$_2$ move apart by climb motion through absorption and emission of self-interstitials. In this case, the partial **b**$_2$ climbs by emission of SI's whereas the partial **b**$_1$ does by absorption. Then, local concentrations of self-interstitials become excess and deficit around the partials **b**$_2$ and **b**$_1$, respectively, by which osmotic forces exert to the partials in the opposite directions of the driving force $f(r)$. As a result, the climb motion is limited by the diffusion of self-interstitials from **b**$_2$ to **b**$_1$.

### 3.2. Diffusive Flow of Self-Interstitials between Two Partial Dislocations

To calculate the diffusive flow rate, the two partials **b**$_1$ and **b**$_2$ are replaced by two cylinders with diameters $d_1$ and $d_2$ as shown in Figure 7b. The cylinders are buried in infinite medium (ice) with an equilibrium concentration $C_e$ (number of self-interstitials/m$^3$) and diffusion coefficient $D_{SI}$ (m$^2$/s) of self-interstitials, resulting in the self-diffusion coefficient $D_{SD} \approx \Omega C_e D_{SI}$ (m$^2$/s) with $\Omega$ (m$^3$) the volume occupied by one molecule in ice. According to [9], an osmotic force $f_{OS}$ (N/m) acts on the two partials due to the local concentration $C$ of self-interstitials as

$$f_{OS} = \frac{k_B T b}{\Omega} \ln\left(\frac{C}{C_e}\right),$$ (13)

where $k_B$ is the Boltzmann's constant, $T$ the absolute temperature, $b$ the Burgers vector length ($c/2$ here), and $C$ the local concentration of self-interstitials around the dislocation core. The local concentration $C_1$ near **b**$_1$ is smaller than $C_e$, and $C_2$ near **b**$_2$ larger than $C_e$. Thus, the osmotic force $f_{OS}$ on the partial **b**$_1$ is toward $+x$ direction, whereas the force exerting on the partial **b**$_2$ is toward $-x$ direction. In steady state, we can assume $f(r) = f_{OS}$, and then obtain the local concentrations as

$$\begin{aligned} C_1 &= C_e \exp\left(-\frac{f(r)\Omega}{k_B T b}\right) \\ C_2 &= C_e \exp\left(\frac{f(r)\Omega}{k_B T b}\right) \end{aligned}$$ (14)

Assuming the concentrations at the cylinder surfaces remain at $C_1$ and $C_2$ as shown in Figure 7b, the flux of self-interstitials per unit thickness of the medium (i.e., per unit length of a cylinder) from the cylinder 2 to 1 at steady state is

$$q = k_{SF} D_{SD}(C_2 - C_1),$$ (15)

where $k_{SF}$ is a shape factor and $D_{SD}$ the self-diffusion coefficient in the medium (ice). According to Incropera et al. [32], the shape factor $k_{SF}$ for unit thickness of the medium depends on the diameter of the cylinders $d = d_1 = d_2$ and the distance $r$ between the two cylinders as

$$k_{SF} = \frac{2\pi}{\cosh^{-1}\left(\frac{2r^2-d^2}{d^2}\right)}. \tag{16}$$

The self-diffusion coefficient is given by $D_{SD} = D_0\exp(-E_{SD}/k_BT)$ with $D_0 = 1.21 \times 10^{-4}$ (m$^2$/s) and $E_{SD} = 0.56$ (eV) [2]. These values of the constants are calculated by the equilibrium concentration $C_e$ (number of self-interstitials/m$^3$) [28] and the diffusion coefficient $D_{SI}$ (m$^2$/s) [29] of self-interstitials separately determined by the methods using X-ray diffraction topography, instead of the diffusion coefficients of isotopes $^3$H and $^{18}$O [31]. As the total number of self-interstitials that have passed through a cross-sectional area of the medium with a thickness $L$ within a unit time equals $qL$ (number of self-interstitials/s), the total volume change caused by migration of self-interstitials within a unit time equals $qL\Omega = V_c(c/2)L$ for the climb velocity $V_c$ (m/s), resulting in $V_c = q\Omega/(c/2)$. Then, the time required to attain a dissociation distance $R$ is calculated by

$$t = \int_{R_0}^{R} \frac{1}{V_c}dr, \tag{17}$$

with $R_0 \geq d$. This treatment has assumed a continuum body (i.e., $d$, $r \gg c$), but to gain a basic understanding of the dissociation-by-climb process, we apply it to the case with $d$ and $r$ comparable to the lattice spacings. Similar to our treatment of the glide case, we use the equations to calculate the dissociation-beginning time $t_b$ and the dissociation-complete time $t_d$. To accommodate uncertainty in the cylinder diameter $d$, we run the calculations for various values of $d$ with initial distance $R_0 = d$ as shown in Table 4.

**Table 4.** Dissociation-beginning time $t_b$ and dissociation-completing time $t_d$ of a non-basal dislocation with Burgers vector <c + a> at temperatures $T$. For the parameters $d$, $R_0$, and $R$, see Equations (12) and (13).

| | Cylinder Diameter $d$ (nm) (=$R_0$) | Final Distance $R$ (nm) | Dissociation-Beginning and -Completing Times $t_b$, $t_d$ (s) | | | |
|---|---|---|---|---|---|---|
| | | | $T$ = 273 (K) (0 °C) | 253 (K) (−20 °C) | 223 (K) (−50 °C) | 183 (K) (−90 °C) |
| $t_d$ | 2.94 (=4c) | 495 (=0.99$w_e$) *1 | $5.5 \times 10^6$ (64 d) | $1.3 \times 10^8$ (4.1 y) | $4.2 \times 10^{10}$ (1.3 Ky) | $1.9 \times 10^{15}$ (61 My) |
| | 1.47 (=2c) | | $6.2 \times 10^6$ (72 d) | $1.4 \times 10^8$ (4.6 y) | $4.8 \times 10^{10}$ (1.5 Ky) | $2.2 \times 10^{15}$ (68 My) |
| $t_b$ | 0.734 (=c) | 1.35 (=3a) | 0.15 | 3.0 | 720 | $1.7 \times 10^7$ (200 d) |
| | 1.47 (=2c) | 2.25 (=5a) | 0.63 | 13.5 | 3840 | $1.3 \times 10^8$ (4.1 y) |

*1 The dissociation-completing time $t_d$ was calculated for $R = 0.99w_e$ instead of $w_e$ to avoid calculation errors arising from a very small difference between $C_1$ and $C_2$ for $R$ very close to $w_e$.

The dissociation-beginning time $t_b$ is uncertain due to its sensitivity to $d$. Nevertheless, the results in Table 4 show that such dissociations along the basal plane take more than one second except close to the melting temperature. As a result, this dissociation may often block glide motion of a non-basal perfect dislocation <c + a> introduced by deformation. For example, perfect dislocations with Burgers vector $c + a_1$ can glide on the prismatic $(01\bar{1}0)$, pyramidal $(1\bar{1}01)$, and the second pyramidal $(2\bar{1}\bar{1}2)$ planes (see Figure 1b), thus allowing continuous non-basal shear deformation until they transform into the sessile dissociated state. This nature of the dislocation <c + a> helps us to understand the non-basal slip deformation in ice [33,34]. On laboratory timescales, such blocking should be almost impossible at temperatures below −90 °C. Above this temperature, the glide motion of dislocations should usually

be blocked by the dissociation. Concerning their extended widths, the dissociation-completing time $t_d$ is as large as several tens of days even at 0 °C, meaning that their widths should be much smaller than their equilibrium value.

In addition, widely extended dislocations of this type cannot nucleate at grain boundaries and free surfaces by deformation. The reason is that, unlike the previous case with glissile extended dislocations, sessile partial dislocations with Burgers vector <**c**/2 + **p**> generally cannot nucleate by deformation because the associated stacking fault can exist only on the basal plane on which the partials cannot glide. Only perfect dislocations with Burgers vector <**c** + **a**> nucleate at grain boundaries and free surfaces by deformation, and they glide on the pyramidal plane until dissociating into <**c**/2 + **p**> on the basal plane. Therefore, we cannot expect widely extended sessile partial dislocations to form by deformation. Instead of deformation, however, partial dislocations <**c**/2 + **p**> associated with widely extended stacking faults can form by segregation or emission of self-interstitials during temperature changes [26–30].

## 4. Summary

Dissociation processes of perfect dislocations in ice have been examined in detail for the first time. We calculated a dissociation-beginning time $t_b$ and the dissociation-completing time $t_d$ for the glissile and sessile extended dislocations with total Burgers vectors <**a**> and <**c** + **a**> for a range of temperatures. Because these dislocations have large equilibrium widths $w_e$, they have long dissociation-completing times $t_d$ at the lower temperatures. For example, for the glissile extended dislocation <**a**>, $t_d$ was found to exceed $10^3$ years below −150 °C, but was very sensitive to temperature, being less than one second above −50 °C. For the sessile extended dislocation <**c** + **a**>, $t_d$ was several tens of days even at 0 °C. Thus, even though their equilibrium widths $w_e$ were calculated to be as large as 500 nm, their measured widths should greatly depend on the temperature history after they have been introduced.

The dissociation-beginning time $t_b$ was introduced to determine whether dislocations would likely be observed in an undissociated or dissociated state. For the perfect dislocation <**a**>, this time was found to be less than one second above −80 °C, but about one year at −150 °C. Thus, at low temperatures, a recently introduced perfect screw dislocations of Burgers vector <**a**> has time to cross-slip to prismatic and pyramidal planes before dissociating, but not at high temperatures associated with the transformation from the shuffle-set to the glide-set dislocations. As a result, the behavior of basal dislocations <**a**> under deformation should depend on temperature. For perfect dislocations with Burgers vector <**c** + **a**>, $t_b$ was found to be larger than one second at all temperatures except close to the melting temperature. Thus, except near melting, such dislocations can glide on prismatic and pyramidal planes, allowing continuous non-basal shear deformation until they dissociate into sessile extended dislocations.

Furthermore, we demonstrated that the extended width of a basal screw dislocation varies with the applied shear stress $\tau$ exerted normal to the dislocation line. The equilibrium width $r_e$ under application of $\tau$ was found to be inversely proportional to the factor $\gamma_p \pm \tau b_p/2$, with $\gamma_p$ the stacking fault energy of the Shockley type. The glide-set dislocations extended on $S$ and $S'$ are oppositely affected by this factor. Unlike the case with most other materials, dislocations in ice can have $\tau b_p/2$ equal or exceed $\gamma_p$. Thus, the equilibrium width can become infinite. In this case, for example, the $S'$-type stacking faults must spread over the entire crystal grain during deformation. In contrast, the $S$-type stacking faults must shrink to much less than $w_e$ under a shear stress $\tau$ larger than $2\gamma_p/b_p$, assisting the cross-slip of basal screw dislocations to prismatic and pyramidal slip planes with transition from the glide set to the shuffle set.

**Funding:** This research received no external funding.

**Conflicts of Interest:** The author declares no conflict of interest.

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
