# Peer review of "Dissociation Behavior of Dislocations in Ice"

_crystals, doi:10.3390/cryst9080386_

Round 1

Reviewer 1 Report

This is a very nice paper examining the dissociation of dislocations in ice. It introduces rate dependence to determine the time needed to start and finish the dislocation dissociation process using traditional thermal activation and examines the rate dependence using a range of possible values taken from the literature. The derivation are based on equations for interaction forces accepted by the community (from Hirth and Lothe). Thus, all modeling and calculations are to my mind correct and leads up to a better understanding of the dislocation dynamics in ice. 

My only concern is related to Figure 1a, which is the basis for stacking faults, perfect and partial dislocations shown in Figures 2 and 3. The figure caption states "Balls (oxygen atoms) are connected by sticks (hydrogen bonds) to form the tetrahedral arrangement of water molecules." This statement is to my mind a bit misleading as the environment around the oxygen atoms can have tetrahedral symmetry - two of the "hydrogen bonds" must have protons that are part of the oxygen forming the water molecule and two of the "hydrogen bonds" must have protons bonded to other oxygen forming two other water molecules - i.e., of the four "hydrogen bonds" that connect each oxygen two bonds must have the protons closed to the oxygen and two must have protons closer to the connecting oxygen. Although all "hydrogen bonds" will be the same the relative position of hydrogen and oxygen in the water molecules will affect how hydrogen bonds can be broken and reformed. the current version of the paper does not address this issue at all but treats all hydrogen bonds as the same by omitting the position of the protons. It is possible that breaking a hydrogen bond and reforming it with another hydrogen bound could result in linkage of two protons or two oxygen, respectively (which could not be hydrogen bonds). Thus, I would like to see a figure showing the real structure of hexagonal ice (including the positions of the hydrogen) as well as some comment on [or illustration of] that the bonds in figures 2 and 3 in fact are true hydrogen bonds (and not H-H or O-O bonds due to the bond-breaking and bond-reformation process).  

Reviewer 2 Report

Title: Dissociation Behavior of Dislocations in Ice

Authors: Takeo Hondoh

Summary: The manuscript discusses the dissociation of dislocations in hexagonal ice, which exist in basal, prismatic and pyramidal planes, into extended partials and derives two time scales: dissociation-beginning and dissociation-completing times. As opposed to other crystals, it is shown that these time scales are extremely sensitive to temperature. Differences in the timescales derived for dislocations in the basal plane and the prismatic/pyramidal planes are highlighted. For instance it is observed that basal dislocations almost always (say at negative 50 degrees Celsius) exist as extended partials limiting their slip to basal plane (no cross slip). On the other hand, the author observes that although prismatic/pyramidal dislocations immediately (within seconds at negative 50C) dissociate, it takes much longer for the dissociation to complete (~1000 years). From this the author infers that non-basal dislocation slip exists when subjected to deformation. The author also notes that for basal slip, one class of stacking faults expand while the other shrinks which is a unique feature of ice as compared to other crystalline metals.

Review:

The topic of the manuscript is quite important for the community and relevant to the journal. Provided the assumptions used in the derivation of timescales are valid, the arguments used to arrive at the conclusion are sound and originate from classical dislocation theory. But the results of the article crucially depend on the assumptions in the derivation of the timescales which I will discuss below. The article is well-written with a clear description of figures and tables. 

In my view, the article is suitable for publication provided the author takes in the consideration the following comments.

The author derives t_b (dissociation-beginning time) assuming that the distance between the two partials is 3*a, where a is the lattice constant. Is the dislocation theory being used valid at such small length scales where the cores interact? There is no comment with regards to this approximation.

The author assumes that the Frank partials observed in ice originate due to the dissociation of perfect dislocations. At least in metals, this is certainly not the case as Frank partials form due to the agglomeration of vacancies or interstitials and need not originate from a perfect dislocation. The author introduces the diffusion of interstitials to describe the phenomenon of a perfect prismatic dislocation dissociating into Frank partials but does not cite any experimental or simulation results where this has been conclusively observed. In the absence of a citation, I think the model presented in the manuscript for non-basal extended dislocation is not validated enough.

In section 1.3, the author states that the dominant slip system on the basal plane - shuffle set vs glide set - is controversial. Having stated that there is a controversy, it is imperative that appropriate citation be included to backup this statement.

The author cites experimental works but does not touch upon any molecular dynamics studies that can easily validate some of the assumptions listed above.

In Fig 7(a), the sum of the Burgers vectors do not point in the direction of c.
